# Identification of PARP-1, Histone H1 and SIRT-1 as New Regulators of Breast Cancer-Related Aromatase Promoter I.3/II

**DOI:** 10.3390/cells9020427

**Published:** 2020-02-12

**Authors:** Alexander Kaiser, Thomas Krüger, Gabriele Eiselt, Joachim Bechler, Olaf Kniemeyer, Otmar Huber, Martin Schmidt

**Affiliations:** 1Institute for Biochemistry II, Jena University Hospital, Friedrich Schiller University, 07743 Jena, Germany; alexander.kaiser@med.uni-jena.de (A.K.); gabriele.eiselt@med.uni-jena.de (G.E.); otmar.huber@med.uni-jena.de (O.H.); 2Department of Internal Medicine II, Jena University Hospital, Friedrich Schiller University, 07743 Jena, Germany; 3Leibniz Institute for Natural Product Research and Infection Biology, Hans Knöll Institute (HKI), 07745 Jena, Germany; thomas.krueger@leibniz-hki.de (T.K.); olaf.kniemeyer@leibniz-hki.de (O.K.); 4Department of Gynecology and Obstetrics, Robert-Koch-Hospital, 99510 Apolda, Germany; j.bechler@rkk-apolda.de

**Keywords:** breast cancer, estrogen synthesis, breast adipose fibroblasts, adipose stromal cells, epigenetics, SIRT-1, PARP-1

## Abstract

Paracrine interactions between malignant estrogen receptor positive (ER^+^) breast cancer cells and breast adipose fibroblasts (BAFs) stimulate estrogen biosynthesis by aromatase in BAFs. In breast cancer, mainly the cAMP-responsive promoter I.3/II-region mediates excessive aromatase expression. A rare single nucleotide variant (SNV) in this promoter region, which caused 70% reduction in promoter activity, was utilized for the identification of novel regulators of aromatase expression. To this end, normal and mutant promoter activities were measured in luciferase reporter gene assays. DNA-binding proteins were captured by DNA-affinity and identified by mass spectrometry. The DNA binding of proteins was analyzed using electrophoretic mobility shift assays, immunoprecipitation-based in vitro binding assays and by chromatin immunoprecipitation in BAFs in vivo. Protein expression and parylation were analyzed by western blotting. Aromatase activities and RNA-expression were measured in BAFs. Functional consequences of poly (ADP-ribose) polymerase-1 (PARP-1) knock-out, rescue or overexpression, respectively, were analyzed in murine embryonic fibroblasts (MEFs) and the 3T3-L1 cell model. In summary, PARP-1 and histone H1 (H1) were identified as critical regulators of aromatase expression. PARP-1-binding to the SNV-region was crucial for aromatase promoter activation. PARP-1 parylated H1 and competed with H1 for DNA-binding, thereby inhibiting its gene silencing action. In MEFs (PARP-1 knock-out and wild-type) and BAFs, PARP-1-mediated induction of the aromatase promoter showed bi-phasic dose responses in overexpression and inhibitor experiments, respectively. The HDAC-inhibitors butyrate, panobinostat and selisistat enhanced promoter I.3/II-mediated gene expression dependent on PARP-1-activity. Forskolin stimulation of BAFs increased promoter I.3/II-occupancy by PARP-1, whereas SIRT-1 competed with PARP-1 for DNA binding but independently activated the promoter I.3/II. Consistently, the inhibition of both PARP-1 and SIRT-1 increased the NAD^+^/NADH-ratio in BAFs. This suggests that cellular NAD^+^/NADH ratios control the complex interactions of PARP-1, H1 and SIRT-1 and regulate the interplay of parylation and acetylation/de-acetylation events with low NAD^+^/NADH ratios (reverse Warburg effect), promoting PARP-1 activation and estrogen synthesis in BAFs. Therefore, PARP-1 inhibitors could be useful in the treatment of estrogen-dependent breast cancers.

## 1. Introduction

Breast cancers are the cancers with the highest incidence in the female population in both the United States and Germany [1,2]. Up to 50–80% of all breast cancers are estrogen receptor-positive (ER^+^) [3]. Excessive amounts of adipose tissue are associated with a higher risk of breast cancer and cells from adipose tissue release various factors supporting tumor growth [4]. A direct link exists between malignant cells in ER^+^ breast cancers and breast adipose fibroblasts (BAFs) representing cancer-associated fibroblasts (CAFs) derived from preadipocytes: namely, that paracrine interactions of tumor cells and BAFs are essential for estrogen biosynthesis by the enzyme cytochrome P-450 aromatase (gene *CYP19A1*, chromosome 15q21.2) in BAFs [5,6]. The expression of the aromatase gene is controlled by a tissue-dependent promoter system. In cancer-associated BAFs, promoter usage is altered, with promoters I.3 and II, normally not used in BAFs, driving increased transcription [7]. These promoters are activated by tumor-derived factors, including prostaglandin E_2_ (PGE_2_), via cAMP-dependent and cAMP-independent signaling mechanisms [8,9]. Thus, tumor-induced aromatase expression and estrogen biosynthesis in BAFs promote ER^+^ tumor growth. When ER^+^ tumors become resistant to endocrine therapies, a declined long-term outcome has been observed [10]. Therefore, despite there being a large body of knowledge about the regulation of aromatase in relation to breast cancer, there is still a need for new targetable mechanisms involved in the regulation of estrogen synthesis and action [11].

The enzyme poly (ADP-ribose) polymerase 1 (PARP-1, gene *PARP1*) catalyzes the poly-ADP-ribosylation (parylation) of target proteins through a NAD^+^-dependent mechanism. It recognizes various substrate proteins via different binding motives [12] and is involved in cellular processes related to DNA methylation, the regulation of splicing, chromatin modulation, DNA repair and transcriptional co-regulation such as histone modifications. Furthermore, the histone deacetylase (HDAC) sirtuin 1 (SIRT-1, gene *SIRT1*) and PARP-1 are partial antagonists in several functional contexts due to their competition for NAD^+^, e.g., in response to DNA damage [13]. Presently, PARP-1 inhibition is mainly utilized in the treatment of triple-negative breast cancers (TNBCs) [14], but not in the treatment of ER^+^ breast cancers.

A potential relevance of PARP-1 inhibition in ER^+^ breast cancers might emerge from our functional analyses of a new single nucleotide variant (SNV) within the aromatase promoter I.3/II-region (SNV(T-241C); NC_000015.10:n.51243270T>C; GRCh38.p13 human genome reference). This SNV leads to a 70% decrease in aromatase promoter I.3/II-mediated transcription. DNA-affinity purification and mass spectrometry identified PARP-1 and histone H1 (genes *HIST1H1A-E*) as alternating binders of this SNV region. A dose-dependent effect of PARP-1 as an enhancer of aromatase transcription was established utilizing PARP-1-knockout MEFs and PARP-1 overexpression. Furthermore, a functional interaction of PARP-1 and SIRT-1 was demonstrated through specific inhibitors and chromatin immunoprecipitation (ChIP) in human BAFs. Thus, PARP-1 may be a potential new target in estrogen-dependent breast tumors.

## 2. Materials and Methods

All chemicals used were of analytical or cell culture grade. All oligonucleotides were from Metabion (Steinkirchen, Germany).

### 2.1. Cells and Cell Culture

3T3-L1 cells were obtained from ATCC (Manassas, VA, USA). The cells were cultured in Dulbecco’s modified Eagle’s medium (DMEM) (Sigma, Taufkirchen, Germany) containing 10% (*v*/*v*) fetal bovine serum (FBS) (PAN-Biotech, Aidenbach, Germany) and 40 µg/mL gentamicin. PARP-1 knock-out and wild-type (wt) murine embryonic fibroblasts (MEFs) were kindly provided by Z. Wang [15]. The cells were cultured in DMEM containing 10% (*v*/*v*) FBS, 40 µg/mL gentamicin, 1 mM sodium pyruvate, 2 mM l-glutamine and 25 mM HEPES. Cell lines were obtained from the original sources (ATCC, producer) and therefore were not reauthenticated. They were used for ten passages. Each stock was tested negative for mycoplasma. Human BAFs were isolated from adipose tissue of healthy patients undergoing cosmetic breast surgery. Patients gave informed written consent according to a protocol approved by the ethics committee of the Jena University Hospital (Ref.-Nr. 4285-12/14). BAFs were isolated and cultured in medium 199 containing 10% (*v*/*v*) FBS, and 40 µg/mL gentamicin, as described previously [16]. Confluent primary human BAFs, resembling almost exclusively preadipocytes, were subcultured only once. All cultured cells were maintained at 37 °C in a humidified atmosphere with 5% CO_2_ and 95% air content, except for T3-L1-preadipocytes, which were grown in 7.5% CO_2_ and 92.5% air.

All cell stimulations or inhibitions were performed for 24 h in a serum-free medium as described previously [16]. Aromatase promoter I.3 and II was pharmacologically activated by 10 µM forskolin (Cayman Chemicals, Ann Arbor, MI, USA) [6]. Furthermore, the cells were treated with PARP-1 inhibitor PJ34 (Selleck Chemicals S7300, Houston, TX, USA), HDAC class I/IIa inhibitor n-butyrate (Sigma B5887, Taufkirchen, Germany), HDAC class I/II/IV inhibitor Panobinostat (LBH589, Selleck Chemicals S1030) and SIRT-1 inhibitor selisistat (EX527, Selleck Chemicals S1541). Where appropriate, DMSO and ethanol solvent controls were carried out in parallel.

### 2.2. Aromatase Activity Testing

The in vivo evaluation of aromatase function in BAFs was performed by the tritium water release assay in 24-well plates as described previously [16,17]. All conditions were tested in triplicate per experiment.

### 2.3. NAD^+^/NADH Quantification

NAD^+^/NADH in BAFs was quantified using the colorimetric EZScreen^TM^ NAD^+^/NADH Assay Kit (BioVision; Milpitas, CA, USA) according to manufacturer’s protocol. The NAD^+^ and NADH extraction protocol was adapted from a published method [18].

### 2.4. Preparation of Soluble Nuclear Extracts

Preparation of soluble nuclear extracts was based on a method published by Wilde et al. [19]. The protein concentration was quantified by the Bradford method [20].

### 2.5. Isolation and Identification of Putative DNA-Binding Proteins

Putative aromatase promoter I.3/II-region binding proteins in soluble nuclear extracts from 3T3-L1 cells were purified using biotinylated oligonucleotides bound to M-280 streptavidin magnetic dynabeads^®^ (Invitrogen; Carlsbad, CA, USA) according to the manufacturer’s protocol. Magnetic beads (48 µL) in wash buffer I (2 M NaCl, 1 mM EDTA, 10 mM Tris/HCl, pH 7.5) were incubated with 1200 pmol 5′-biotinylated normal sequence oligonucleotide (Appendix B, Table A1) for 20 min under rotation. After 3 washing steps in wash buffer I, oligonucleotide-coupled beads were mixed with 1800 µg soluble nuclear extract proteins (720 µL), 192 µL 10-fold binding buffer (1 M NaCl, 1 mM EDTA, 0.5 M Tris/HCl, pH 7.5), 940.8 µL water and 19.2 µL poly-dI/dC (10 µg/µL), and incubated for 30 min under rotation at room temperature. After 3 washing steps in two sample volumes each of wash buffer II (10% (*v*/*v*) 10-fold binding buffer, 37.5% (*v*/*v*) nuclear extraction buffer (20 mM HEPES pH 7.9, 400 mM NaCl, 1 mM EDTA, 1 mM EGTA, 1 mM dithiothreitol, 1 mM PMSF), 52.5% (*v*/*v*) water), protein-binding magnetic beads were resuspended in 20 µL 1.5-fold Laemmli sample buffer (93.75 mM Tris/HCl pH 6.8, 3% (*w*/*v*) SDS, 15% (*v*/*v*) glycerin, 0.015% (*w*/*v*) bromphenol blue) without 2-mercaptoethanol. Thereafter, 0.5 mg/mL dithiothreitol (incubation 3 min, 95 °C) and 2 mg/mL iodoacetamide (incubation 30 min, 37 °C) were added.

The captured proteins were separated on 8% SDS-polyacrylamide gels [21], which were stained with Coomassie-blue [22] or ruthenium-red [23], respectively. Subsequent proteomic analysis was basically performed as described by Baldin et al. [24]. Specific oligonucleotide-binding proteins were excised and digested with trypsin, as described in Shevchenko et al. [25]. Extracted peptides (acetonitril/0.1% (*v*/*v*) trifluoroacetic acid; 1:1) were analyzed by MALDI-TOF MS and Nano-ESI-Quadrupol-TOF MS.

MALDI-TOF MS: Extracted peptides were mixed with a saturated α-cyano-4-hydroxycinnamic acid solution (dissolved in acetonitril), allowed to dry on a stainless-steel anchor chip target and subsequently analyzed by MALDI-TOF-TOF (ultrafleXtreme; Bruker Daltonics, Bremen, Germany).

Nano-ESI-Quadrupol-TOF MS: The chromatographic separation of extracted peptides was performed on an Acclaim Pep MAP RSLC column in an Ultimate 3000 nano RSLC Systems (Thermo Fisher Scientific, Dreieich, Germany) (eluent A: 0.1% (*v*/*v*) formic acid in water; eluent B: 0.1% (*v*/*v*) formic acid in a 90% (*v*/*v*) acetonitril/10% (*v*/*v*) water mix); gradient: 0–4 min 4% (*v*/*v*) eluent B, 4–36 min to 35% (*v*/*v*) eluent B, 36–40 min to 50% (*v*/*v*) eluent B, 40–41 min to 96% (*v*/*v*) eluent B, 41–45 min 96% (*v*/*v*) eluent B, 45–60 min 4% (*v*/*v*) eluent B. MS was done in first-generation microTOF-Q MS (Bruker Daltonics, Bremen, Germany). The MS peaks were identified by searching the NCBI-database using the MASCOT interface (MASCOT 2.1.03, Matrix Science, London, UK) with the following parameters: Cys as S-carbamidomethyl-derivative and Met in oxidized form (variable), one missed cleavage-site, peptide mass tolerance of 300 ppm (MALDI-TOF MS) or 100 ppm (Nano-ESI-Quadrupol-TOF MS). Hits were considered significant according to the MASCOT score (*p* = 0.05). The database research was improved by iterative recalibration and application of the peak rejection algorithm filter of the Score Booster tool implemented into the Proteinscape 3.0 database software (Protagen Dortmund, Germany).

### 2.6. Electrophoretic Mobility Shift Assays

For electrophoretic mobility shift assays (EMSA), 10 µg soluble nuclear extract protein per condition was incubated in the presence of binding buffer (50 mM Tris/HCl pH 7.5, 0.1 M NaCl, 0.1 mM EDTA, 5 mM 2-mercaptoethanol) for 30 min at 37 °C with various double-stranded probes (Appendix B, Table A1)—25 pmol of a Cy5-labeled normal sequence probe (either alone or in the presence of a 20-fold molar excess of an unlabeled normal sequence probe (competitor)), or 25 pmol of a Cy5-labeled SNV-containing probe or Cy5-labeled quadruple mutation probe (complete destruction of putative binding-sites). For antibody competition, 2 µL of anti-PARP-1 antibodies (Appendix B, Table A2) were incubated for 30 min before the addition of probes. Separations were carried out on a 6% non-denaturing acrylamide gel at 4 °C (18 cm, 300 V, and 70 min; [26]). The wet gels were directly scanned on a Fuji FLA-3000 imaging system and quantified using the AIDA Software (Raytest, Straubenhardt, Germany).

### 2.7. Immunoprecipitation-Based DNA-Binding Assay

An immunoprecipitation-based DNA-binding assay protocol was developed for PARP-1 and histone H1, respectively. Soluble nuclear extract proteins (50 µg) were pre-incubated with 2 µL pre-cleared (in soluble nuclear extract buffer) Protein G-Sepharose 4 Fast Flow (GE Healthcare, Freiburg, Germany) at 4 °C in a rotator to eliminate proteins binding non-specifically to protein G. After centrifugation of the pre-incubated samples (20 s, 12,000× *g*, 4 °C), the supernatants were transferred into new tubes and incubated with either 1 µL anti-histone H1 or 1 µL anti-PARP-1 antibodies (Appendix B, Table A2) for 24 h at 4 °C under constant rotation. After the antibody incubation, 20 µL pre-cleared Protein G-Sepharose 4 Fast Flow was added and incubated for 4 h at 4 °C under constant rotation. For the final isolation of anti-histone H1 or anti-PARP-1 immunoprecipitates, respectively, the samples were washed three times in a three-fold volume of DNA-binding buffer C (20 mM HEPES pH 7.9, 1 mM EDTA, 1 mM EGTA, 1 mM DTT, 1 mM PMSF) with centrifugation after each step (20 s, 12,000× *g*, 4 °C). The final immunoprecipitates were resuspended in 8 µL buffer C.

The DNA-binding reaction was a modification of the sample preparation protocol for electrophoretic mobility shift assays, as described by Taylor et al. [26]. Immunoprecipitates in buffer C (8 µL) were mixed with 1.7 µL 10-fold binding buffer (500 mM Tris/HCl pH 7.5, 1 M NaCl, 1 mM EDTA, 50 mM β-mercaptoethanol) and were incubated with 50 pmol fluorescence-labeled oligonucleotides with normal sequence (either alone or in the presence of a 26-fold molar excess of an unlabeled normal sequence competitor) or sequence with SNV (Appendix B, Table A1). For normal binding reactions, the premix was added to 2 µL Cy5-labeled double-stranded oligonucleotides (25 pmol/µL) in 5.3 µL water. For binding reactions in the presence of a competitor, 8 µL immunoprecipitate, 2 µL fluorescence-labeled double-stranded oligonucleotides and 5.3 µL unlabeled competitor double-stranded oligonucleotides (250 pmol/µL; same sequences as fluorescence-labeled oligonucleotides) were mixed. After incubation for 30 min with rotation at room temperature in the dark, the samples were washed three times in a three-fold sample volume of wash buffer (50% buffer C, 10% 10-fold binding buffer, 40% water), followed by a 20 s centrifugation at 12,000× *g* at room temperature. Finally, the oligonucleotide-bound immunoprecipitates were resuspended in 17 µL wash buffer and transferred to a well of a 96-well plate for fluorescence measurement (excitation 600 nm; emission 670 nm, cut off 630 nm). As a control, the unspecific binding of fluorescent oligonucleotides to Protein G-Sepharose 4 Fast Flow beads treated as described above in the absence of antibodies was analyzed, resulting in negligible fluorescence signals. All conditions were tested in triplicate per experiment.

### 2.8. Western Blotting

Precipitated proteins were separated on 10% SDS-polyacrylamide gels [21]. Proteins were transferred onto PVDF membranes using semi-dry blotting at 0.8 mA/cm^2^ for 40 min [27]. After blocking in WP-T buffer (10 mM Tris/HCl pH 7.5, 100 mM NaCl, 0.1% (*v*/*v*) Tween 20) with 5% (*w*/*v*) skimmed milk powder, the membrane was incubated overnight with the primary antibodies (1:1,000 each; Appendix B, Table A2). After washing in WP-T buffer and further blocking in WP-T buffer with 5% (*w*/*v*) milk powder, the appropriate HRP-conjugated secondary antibodies were added (1:5,000, goat anti-mouse or goat anti-rabbit, Santa Cruz Biotechnology, Dallas, TX, USA). The proteins were detected using enhanced chemiluminescence.

### 2.9. Transfection and Luciferase Reporter Gene Assays

Half-confluent 3T3-L1 preadipocytes or PARP-1 knock-out or wt MEFs in 24-well plates were transfected using Roti-Fect Plus (Carl Roth, Karlsruhe, Germany) according to the manufacturer’s instructions and stimulated 24 h later. To quantify promoter activities, pGL3-basic firefly luciferase plasmid containing aromatase promoter I.3/II with either a normal or SNV sequence were used (Appendix B, Table A3). For PARP-1 overexpression, pSG9M-PARP-1 plasmid was used [28]. Furthermore, we utilized pRL-SV40 plasmid-expressing Renilla luciferase for normalization. All conditions were tested in triplicate per experiment. The luciferase activity measurement was described by Hampf and Gossen [29].

### 2.10. Chromatin Immunoprecipitation (ChIP)

The chromatin immunoprecipitation protocol is a modified version of that published by Weiske and Huber [30]. BAFs from four 10 cm dishes per condition were used per experiment. For each Protein G-based immunoprecipitation, 1 µg anti-histone H1, anti-PARP-1 or anti-SIRT-1 antibody was used per 5 µg of cross-linked DNA, respectively (Appendix B, Table A2). PCR was carried out using Paq5000 DNA-polymerase (Agilent Technologies, Waldbronn, Germany) in a Veriti^®^ 96-Well Thermal Cycler (Applied Biosystems, Darmstadt, Germany). Two SNV region-spanning primer sets were used (Appendix A, Appendix A). The products were analyzed on 12% polyacrylamide gels stained with ethidium bromide, as described previously [19].

### 2.11. Quantification of Aromatase mRNA-Expression in BAFs

The quantification of full-length aromatase mRNA-expression and utilization of promoters I.3 and II, respectively, was performed by quantitative real-time PCR (qRT-PCR), as described in detail by Wilde et al. [19]. Primer sequences are given in the Appendix A, Appendix A. All conditions were tested in triplicate per experiment. The evaluation of the PCR results was done by the ΔΔC_T_-method [31].

### 2.12. Quantification of Aromatase Promoter-Utilization in Transfected PARP-1 Wild-Type and Knock-Out MEFs

Murine PARP-1 wild-type and knock-out MEFs were transfected using pGL3-PII-522 wt plasmid, as described (Appendix B, Table A3). After 24 h of forskolin stimulation, DNA and mRNA were isolated using the AllPrep DNA/RNA Mini Kit (Qiagen; Hilden, Germany) with on-column DNA-digestion during the RNA-isolation. cDNA was synthesized as described [19]. The ABsolute SYBR Green Rox Mix (Thermo Scientific; Schwerte, Germany) was used for qRT-PCR in a StepOnePlus™ real-time PCR system (Applied Biosystems). The transfected aromatase promoter-dependent expression of luciferase or the expected fusion sequences containing 5′-ends from aromatase exon I.3 (derived from the 5′- or 3′-region of exon I.3, respectively) and a 3′-ends derived from luciferase were measured using the primers described in the Appendix A, Appendix A, and cDNA as templates. Furthermore, the amounts of firefly luciferase in relation to murine glyceraldehyde-3-phosphate dehydrogenase (GAPDH) were measured using isolated DNA from the same cells as the templates, enabling a correction for transfection efficiencies. All conditions were tested in triplicate per experiment. The evaluation of the PCR results was performed by an accordingly modified ΔΔC_T_-method [31].

### 2.13. Statistical Analyses

Statistical analyses of all experiments and creation of diagrams were carried out using SigmaPlot 13 software (Systat Software GmbH, Erkrath, Germany). The data are presented as means ± sem or using box plots, were appropriate. Initial normal distribution of values was tested by the method of Shapiro-Wilk. Normally distributed values were compared to another group by Student´s *t*-test or by one-way ANOVA for multiple comparisons. In the case of non-normally distributed values, two groups were compared by the Mann–Whitney U-test. For all tests, the significance criterion *p* < 0.05 was used.

## 3. Results

### 3.1. SNV-Dependent Protein Complex Formation in the Aromatase Promoter I.3/II Region

We identified a new, extremely rare single nucleotide variant (SNV) in the aromatase promoter I.3/II-region of a healthy DNA-donor (SNV(T-241C); NC_000015.10:n.51243270T>C; GRCh38.p7 human genome reference; Appendix A, Appendix A). This SNV decreased aromatase promoter I.3/II activity in luciferase-reporter gene assays in 3T3-L1 cells by up to 70%, when the cells were stimulated with the cAMP-elevating agonists di-butyryl-cAMP or forskolin (Figure 1A). This indicates a crucial role for the base-pair at position −241 in relation to the transcriptional start site (TSS) of aromatase promoter II. Two specific protein–oligonucleotide complexes could be identified in soluble nuclear extracts from 3T3-L1 preadipocytes in EMSAs using normal and SNV-containing oligonucleotides, respectively (Figure 1B). Complex formation was independent of forskolin, which induces an increase in cAMP and thereby mimics cancer-related aromatase promoter I.3/II activation. Furthermore, complex 1 was only formed with wild-type oligonucleotide, but not on the SNV-containing oligonucleotide or an extended quadruple mutation thereof. Complex 2 was not affected by the SNV. This indicates that the formation of complex 1 might be necessary for the full induction of aromatase transcription.

### 3.2. PARP-1 and Histone H1-Isoforms Bind to the SNV-Region

We subsequently identified the DNA-binding protein(s) of the SNV-dependent complex 1. Firstly, DNA-binding proteins from nuclear extracts of 3T3-L1 cells were purified with SNV sequence-containing oligonucleotides coupled to magnetic beads. After SDS-PAGE, a 110 kDa protein was detectable and identified by mass spectrometry as PARP-1 (Figure 2A and Appendix A, Appendix A). The formation of complex 1 was blocked by anti-PARP-1 antibody in EMSA with soluble nuclear extracts of 3T3-L1 cells, confirming PARP-1 as a component of complex 1 (Figure 2B). Similar experiments were performed with nuclear extracts from PARP-1 wild-type and knock-out MEFs to confirm this observation (Figure 2C,D). In PARP-1 wild-type MEFs, the formation of complex 1 on wild-type oligonucleotide was inhibited by anti-PARP-1 antibody or when a mutated probe was used, confirming the results from the 3T3-L1 cells described above. Furthermore, PARP-1-containing complex 1 (but not complex 2) was massively reduced in PARP-1 knock-out MEFs.

In addition to PARP-1, a second protein band of 32 kDa, specifically binding to the aromatase promoter I.3/II-region, was detectable and identified by mass spectrometry as histone H1-isoforms (Figure 2A and Appendix A, Appendix A).

### 3.3. PARP-1 and Histone H1 Compete for Binding to the Aromatase Promoter I.3/II-Region

In competition experiments using biotin-labeled oligonucleotides for pull-down of associated proteins from 3T3-L1-cell nuclear extracts with magnetic beads, PARP-1 was detectable as the dominant binding protein in ruthenium-stained gels and on western blots (Figure 3A). The addition of a 10-fold molar excess of non-biotinylated competitor oligonucleotide displaced PARP-1 and caused a prominent histone H1-binding. This asymmetrical competition suggests that PARP-1 is a high-affinity/low-concentration binder of the aromatase promoter I.3/II-region, whereas histone H1 represents a low-affinity/high-concentration binder.

Immunoprecipitation-coupled DNA-binding assays using 3T3-L1 nuclear extracts accessorily revealed the involvement of PARP-1 (Figure 3B). Immunoprecipitated PARP-1 revealed a similarly weak binding to an SNV-containing probe (1m), as it did to a wild-type probe in the presence of competitor oligonucleotides. In contrast, immunoprecipitated histone H1 revealed an even greater binding to the SNV-containing probe (1m).

### 3.4. Histone H1 Is Parylated

The molecular weight of unmodified murine histone H1 is 21.79 kDa (uniProtKb database). Nonetheless, histone H1 pulled-down with oligonucleotide-coupled magnetic beads from 3T3-L1 nuclear extracts revealed an apparent molecular mass of 32 kDa (Figure 3A). Western blot analysis of immunoprecipitated histone H1 with a parylation-specific antibody detected a parylated 32 kDa-band (Figure 3C). This observation suggests that PARP-1-catalyzed parylation caused the observed mass shift of histone H1.

### 3.5. A Dual Role for PARP-1 on the Aromatase Promoter I.3/II

Having identified that PARP-1 preferentially binds to the wild-type sequence in the SNV-region, we subsequently endeavored to elucidate the functional consequence of this interaction on the aromatase promoter I.3/II activity. The overexpression of PARP-1 in 3T3-L1 cells resulted in increased firefly luciferase activity, which was even more prominent after forskolin stimulation (Figure 4A). Similar results were observed in PARP-1 knock-out MEFs, where PARP-1 overexpression led to biphasic dose responses (Figure 4B). The maxima of luciferase activities were measured when 150 ng PARP-1 expression plasmid per well was used for transfection and analysis was performed with the reporter gene plasmid containing the wild-type aromatase promoter I.3/II sequence and 75 ng/well using the SNV-sequence. A further increase in PARP-1 expression plasmid resulted in significantly lower reporter gene activity.

Of note, however, is that in the dual-reporter gene assays, PARP-1 overexpression also massively induced the Renilla luciferase vector (Appendix C, Figure A1A). Therefore, we validated and confirmed the inducing effect of PARP-1 on the aromatase promoter region by another approach. PARP-1 knock-out or wild-type MEFs were transfected with the reporter plasmid containing the normal aromatase promoter I.3/II sequence (pGL3-PII-522 wt) and analyzed by RT-qPCR for transcribed mRNAs, which could be normalized exactly via qPCR for the transfected plasmid-DNA. Three amplicons (firefly luciferase-specific, exon I.3-5′-region-specific and exon I.3-3′-region-specific) were tested and revealed a massive 15–40-fold inducing effect of PARP-1 on promoter I.3/II-dependent aromatase transcription (Figure 4C). Taken together, PARP-1 binding clearly affects the aromatase promoter I.3/II-region.

Based on these observations, we hypothesized that the inhibition of PARP-1-activity should inhibit aromatase promoter activity. Surprisingly, we observed that treatment with the PARP-1-inhibitor PJ34 resulted in a dose-dependent increase in reporter gene activities of the firefly luciferase promoter I.3/II wild-type-construct (Figure 4D) and of the Renilla-vector (Appendix B, Figure A1B) in 3T3-L1 cells. This suggests that PARP-1 may also possess inhibitory potential. In contrast, total aromatase expression regulated by promoters I.3 and II as well as aromatase enzyme activity were inhibited by PJ34 in biphasic dose-responses in BAFs (Figure 4E–H). The strongest inhibition was observed with 5–7 µM PJ34, whereas higher concentrations almost restored these values to those of untreated cells. Taken together, these data suggested a dose- and cell type-dependent dual role of PARP-1 in the regulation of aromatase promoter I.3/II activity. Moreover, both of the transcriptional start sites within the promoter I.3/II-region are affected [7].

### 3.6. HDACs Modify Aromatase Promoter I.3/II Activity

Deacetylation is known to affect PARP-1 and histone H1 function [32,33]. Therefore, HDAC inhibitors were screened in aromatase promoter I.3/II reporter gene assays in 3T3-L1 cells. In forskolin-stimulated cells, the HDAC class I/IIa inhibitor N-butyrate increased luciferase activity, which was significantly reduced by PJ34-mediated PARP-1 inhibition (Figure 5A), contrary to the effect of PJ34-treatment in the absence of HDAC inhibitor (see Figure 4D), pointing to an interdependent regulation. The class I/II/IV inhibitor panobinostat increased the aromatase promoter I.3/II driven luciferase activity in both the wild-type and SNV-promoter genotypes (Figure 5B). The SIRT-1 (HDAC class III) inhibitor selisistat, like n-butyrate, augmented luciferase activity, which was significantly reduced by PARP-1 inhibition (Figure 5C). In contrast to panobinostat, the effect of selisistat could only be observed with the normal sequence reporter, which is indicative of an antagonistic participation of SIRT-1 in the control of the aromatase promoter I.3/II SNV-region by PARP-1.

Having identified selisistat/SIRT-1 as the most likely partner for PARP-1, we went back to BAFs. Selisistat significantly inhibited total and promoter I.3/II-specific aromatase mRNA-expression (Figure 5D) as well as aromatase activity in forskolin-treated BAFs (Figure 5E). Interestingly, we observed a statistically not significant tendency for a biphasic dose response to PARP-1 inhibition in the presence of selisistat in the RNA expression measurements, which was diametrically opposite to the U-shaped PJ34 dose response of the controls. The large variances are due to widely differing sensitivities of BAFs to selisistat. In the aromatase activity assays, these effects were not detectable, probably due to interference with post-transcriptional effects of the inhibitors. In summary, SIRT-1 and PARP-1 functionally interact in the control of aromatase expression.

Enzymatic activities of both PARP-1 and SIRT-1 depend on NAD^+^-consumption [13]. Despite the presence of multiple other NAD^+^-metabolizing enzymes, the inhibition of PARP-1 or SIRT-1 or both together resulted in increased NAD^+^/NADH ratios (Figure 5F), indicating that both enzymes consume quite large amounts of, and compete at least locally for, NAD^+^.

### 3.7. Aromatase Expression Is Regulated by an Antagonizing System Including PARP-1, Histone H1 and SIRT-1

To conclusively verify the in vivo binding of PARP-1, histone H1 and SIRT-1 to the SNV-region in the aromatase promoter I.3/II, we performed ChIP assays utilizing control and forskolin-stimulated BAFs. Two overlapping primer sets covering the SNV-containing region of promoter I.3/II were used (Figure 6A). In accordance with the results of the in vitro studies as described above, forskolin treatment lead to a significantly increased PARP-1 binding, when analyzed with primer set 1 (Figure 6B,C). The switch in promoter occupancy was even more obvious when ratios of PARP-1/SIRT-1 and of PARP-1/histone H1 were analyzed (Figure 6D). Moreover, primer set 2 suggested that the partial displacement of histone H1 and SIRT-1 by PARP-1 takes place in the immediate neighborhood of the SNV-region (Figure 6E,F). Here too, the ratios of PARP-1/SIRT-1 and of PARP-1/histone H1 significantly increased in forskolin-stimulated BAFs (Figure 6G), indicating the partial displacement of SIRT-1 and histone H1 by PARP-1. Taken together, these results indicate that PARP-1, H1, and SIRT-1 directly interact in an aromatase promoter I.3/II-region crucial for the induction in BAFs in the vicinity of breast cancer cells in vivo.

## 4. Discussion

Besides its role in DNA damage response and various other processes, the involvement of PARP-1 in transcriptional control has been established [34,35]. The identification of PARP-1 as a transcriptional regulator of aromatase expression signifies the promoter I.3/II-region of aromatase as one of PARP-1´s many targets [36]. More recently, PARP-1 received attention as a therapeutic target in TNBC and is currently the subject of a series of clinical trials of specific inhibitors [14,37,38], based on its function in DNA damage control. In addition, some authors have concluded that PARP-1 inhibition could be useful in a broader range of breast cancers, including ER^+^ breast cancers (not necessarily mutated in the BRCA DNA repair associated genes) [14,39].

Our study provides evidence that the inhibition of PARP-1 might be a pharmacological possibility for treating ER^+^ breast cancers by targeting transcription. By happenstance, we have identified a new rare SNV, located immediately downstream of the TATA-box of aromatase promoter I.3 (see Figure 7), which strongly reduced reporter gene activity driven by the promoter I.3/II-region. Promoter I.3 is intimately connected with promoter II, both functionally and by proximity (the transcriptional start-site (TSS) of promoter II is only 226 bp downstream of the promoter I.3-TSS) [7,8]. PARP-1 preferentially binds to the normal sequence of the SNV. The reduction in binding to the SNV variant in in vitro binding assays was almost equal to a reduction in promotor activity in reporter gene assays, which strongly suggests that PARP-1 is a critical component for the full induction of the breast cancer-associated promoter I.3/II-region.

In principle, it seems counterintuitive that PARP-1 binding to a DNA site that at least overlaps with the TATA-box core-promoter element should enhance transcription. However, several lines of evidence clarify this supposed discrepancy. First, the binding location is typical for PARP-1, as Krishnakumar et al. [36] identified the most probable PARP-1 accumulation to be about 250 bp upstream of TSSs in their large-scale analysis of promoters, almost perfectly fitting the SNV-spacing of 241 bp upstream of the promoter II TSS. Secondly, the overexpression of PARP-1 induces promoter I.3/II controlled reporter gene activity in cell lines, whereas the inhibition of PARP-1 inhibits aromatase expression and activity in BAFs. Thirdly, Wang et al. [40] found that the 5´-region of the non-coding exon I.3 inhibits the translation of the aromatase protein. The results of our qPCR-validation of the reporter gene experiments in MEFs (see Figure 4C) reveal that the presence of PARP-1 leads to preferential transcription of a 5´-truncated exon I.3, resulting in the omission of this inhibitory region. Taken together, these findings indicate a central role of the SNV-region and PARP-1 for induction of aromatase.

We observed biphasic dose responses of the aromatase expression and enzyme activity of BAFs treated with the PARP-1-inhibitor PJ34. This cannot be explained by a stand-alone action of PARP-1 at the aromatase promoter. Furthermore, the overexpression of PARP-1 in PARP-1 knock-out MEFs also resulted in a biphasic dose response, which might be caused by autoparylation at high concentrations of PARP-1 and concomitant autoinhibition [41]. Taken together, all of the results suggest that PARP-1 must be part of a multifactorial system, in which interactions of different partners modulate the aromatase promoter I.3/II activity (Figure 7).

One important interaction partner is histone H1 (H1), which co-purified with PARP-1 in our assays. H1 even exhibited an increased binding to the mutant SNV sequence in line with the functional association of H1 and PARP-1 reported in the literature [34]. Aubin et al. described euchromatin formation by PARP-1-mediated H1 parylation and its displacement, resulting in gene activation [42]. PARP-1 inhibition reduced gene activity via H1-mediated heterochromatin formation. In this context, such dynamic interactions of H1 and PARP-1 in large-scale analyses were reported by Krishnakumar et al. [36]. Our competition experiments and ChIP analyses confirmed this dynamic regulation for the aromatase promoter I.3/II-region as well. Interestingly, most of the H1 from 3T3-L1 cells was parylated. In Western blots of immunoprecipitated H1, as well as in the DNA-coupled bead assays, we found a size shift in H1 to 32 kDa, which is typical for parylation, as described by Huletsky et al. [43].

The involvement of H1 in aromatase promoter I.3/II regulation also indicates the involvement of HDACs. Besides the modulation of the nucleosome configuration by histone deacetylation, HDACs modify transcription factors [44]. HDAC class I/II/IV inhibition in 3T3-L1 cells increased the aromatase promoter I.3/II activity, while concurrent PARP-1-inhibition reduced its activity, indicating that acetylation events necessary for aromatase transcription need PARP-1 for a maximum effect. Contrarily, PARP-1 is activated through acetylation by p300/CBP, which can be reversed by class I HDACs [32].

Nonetheless, a special role in the functional relationship between HDACs and PARP-1 is assigned to class III HDACs (sirtuins). Their deacetylase function involves a NAD^+^-dependent catalytic mechanism [45] and, in particular, SIRT-1 mediates deacetylation of PARP-1 [46]. In our experiments, SIRT-1 inhibition in 3T3-L1 cells resulted in similar effects as HDAC class I/II/IV inhibition. In contrast, in BAFs SIRT-1 inhibition caused a reduction in aromatase expression and activity. This inhibition tended to be alleviated in the presence of moderate concentrations of a PARP-1 inhibitor. In BAFs, SIRT-1 seems to act as an aromatase promoter I.3/II activator. Taken together, experimental evidence suggests a strong functional relationship between SIRT-1 and PARP-1, which may be modulated in a cell type-specific manner. This functional link appears to be the NAD^+^-requirement of both enzymes [34,47,48]. Furthermore, SIRT-1 and PARP-1 have different kinetic features. The Michaelis constant (K_M_) of PARP-1 is 20–60 µM NAD^+^ [49]. K_M_ of SIRT-1 is 150–200 µM NAD^+^ [50]. Therefore, PARP-1 is more active than SIRT-1 at low NAD^+^-levels, as can be observed, for example, in cardiomyocytes [51]. Conversely, high NAD^+^-levels favor increased SIRT-1 activation, leading to PARP-1 inhibition by deacetylation (and inhibition of gene expression) [46]. In fact, in our ChIP experiments, a shift towards stronger promoter I.3/II occupancy by PARP-1 upon forskolin stimulation could be observed.

Furthermore, with PARP-1 and SIRT-1 being involved in the regulation of aromatase expression, the cellular context becomes important. BAFs in the neighborhood of tumor cells may exhibit a reverse Warburg effect like CAFs [52]. This effect is well described for CAFs in breast cancers [53,54]. The reverse Warburg effect is triggered by metabolites like lactate, released from tumor cells. In stromal cells, their utilization can reduce the NAD^+^/NADH ratio [55]. The subsequently low NAD^+^-level favors PARP-1-activity as compared to SIRT-1-activity.

The U-shaped dose responses to PARP-1 inhibition in BAFs result from an interaction with a partner, which itself is functionally coupled to PARP-1 and stimulates aromatase expression independently. Our observations using SIRT-1 inhibition in BAFs suggest that SIRT-1 is this partner. Active PARP-1 deprives SIRT-1 of NAD^+^, thus preventing its own inactivation by SIRT-1. Furthermore, PARP-1 supports euchromatin formation by H1-parylation (in addition to acetylation by acetyltransferases) contributing to the activation of aromatase promoter I.3/II (Figure 7). Only under conditions of PARP-1 inhibition does the cellular NAD^+^-concentration increase to a sufficiently high level to allow SIRT-1 to stimulate aromatase expression via, as of now, unidentified mechanisms.

## 5. Conclusions

In summary, PARP-1 is a key regulator of the aromatase promoter I.3/II activity, which can be activated in BAFs by metabolic coupling to breast cancer cells. Lack of PARP-1 binding to the promoter I.3/II region or PARP-1 inhibition, respectively, reduces aromatase expression as well as estrogen synthesis in BAFs. Thus, low doses of PARP-1 inhibitors might be potentially useful for estrogen-deprivation of ER^+^ breast cancers.

## Figures and Tables

**Figure 1 cells-09-00427-f001:**
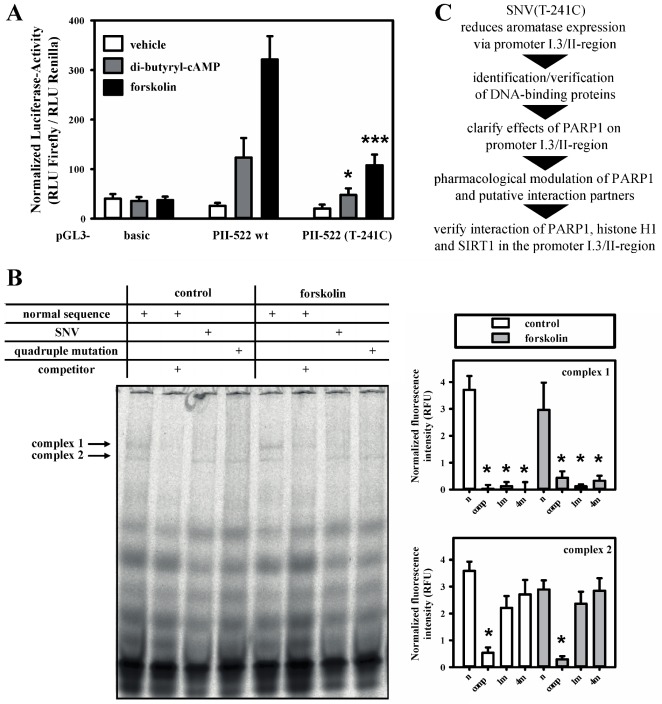
SNV(T-241C) reduces aromatase promoter I.3/II activity and DNA–protein complex formation. (**A**) 3T3-L1 cells were co-transfected with Renilla luciferase pRL-SV40-plasmid for normalization and either pGL3-basic or aromatase promoter I.3/II-containing pGL3-PII-522 wt or pGL3-PII-522 (T-241C) firefly luciferase reporter-plasmids. The cells were treated with vehicle, 1 mM di-butyryl-cAMP or 25 µM forskolin. The data are means of the three experiments done with triplicate replicates of each. Differences caused by SNV(T-241C) versus wild-type were analyzed with Students *t*-test (*n* = 3; * *p* < 0.05; *** *p* < 0.001). (**B**) Soluble nuclear extracts from 3T3-L1 cells were used for electrophoretic mobility shift assays (EMSAs). Complex 1 formation was inhibited by competitor oligonucleotides (comp). SNV (1m)- or quadruple mutation (4m)-containing probes did not form complex 1, as seen with normal sequence probe (n). Complex 2 was solely inhibited by competitor (*n* = 6; one representative experiment shown; * *p* < 0.05 for comparison with normal sequence probe only). (**C**) Flow chart highlighting subsequent steps of investigation (for a more detailed overview see Figure 7).

**Figure 2 cells-09-00427-f002:**
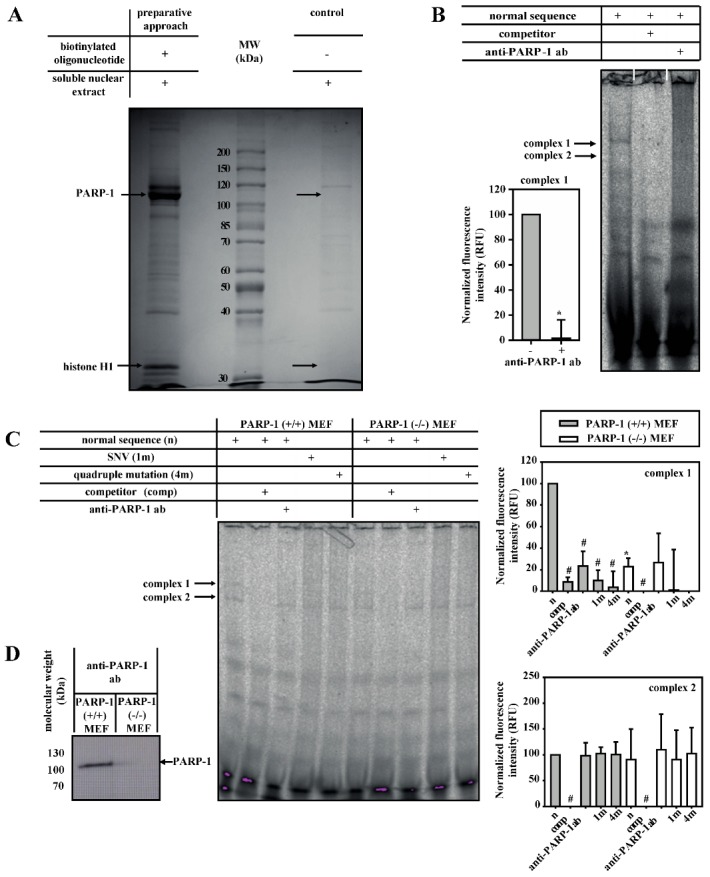
PARP-1 and H1 bind to the wild-type aromatase promoter I.3/II region. (**A**) PARP-1 (110 kDa) and histone H1 (32 kDa) were purified by DNA-affinity pull-down from soluble nuclear extract of 3T3-L1 cells using biotinylated wild-type oligonucleotide. Specifically binding (left) and non-specifically binding (right) proteins were separated by SDS-PAGE and detected by Coomassie Blue staining. Proteins identified by mass spectrometry are indicated by arrows (see also Appendix A, Appendix A). MW, molecular weight. (**B**) Complex 1 from soluble nuclear extracts from 3T3-L1 cells was inhibited by competitor and anti-PARP-1 antibodies (ab) (*n* = 4; * *p* < 0.05). (**C**) Soluble nuclear extracts from PARP-1 wild-type (+/+) MEFs or PARP-1 knock-out (−/−) MEFs were used for EMSAs. Complexes 1 and 2 were inhibited by the competitor, but only complex 1 formation was suppressed when using SNV (1m)- or quadruple mutation (4m)-containing probes. Furthermore, the formation of complex 1 was inhibited by anti-PARP-1 antibodies and was strongly reduced in PARP-1 (−/−) MEFs (*n* = 3; * *p* < 0.05 for (−/−) versus (+/+); # *p* < 0.05 for treatments compared with corresponding normal sequence treatment). (**D**) Western blot detects no PARP-1 in PARP-1 knock-out MEFs.

**Figure 3 cells-09-00427-f003:**
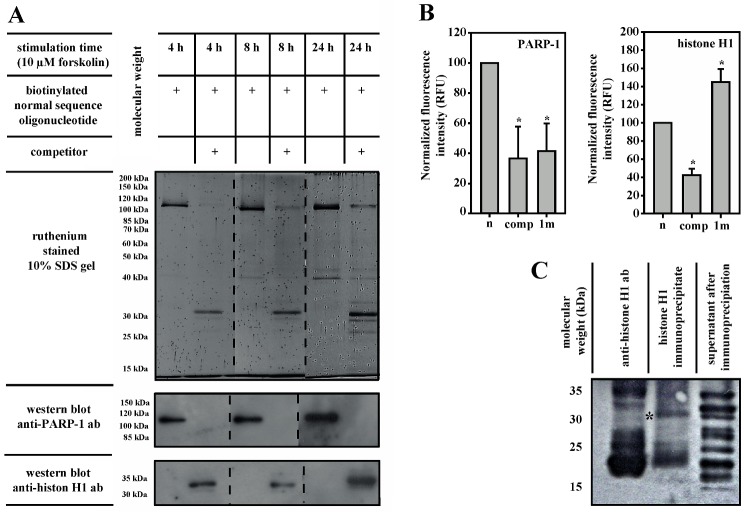
PARP-1 and histone H1 are part of a regulatory system. (**A**) Putative binding proteins in 200 µg soluble nuclear extract of 3T3-L1 cells were purified using magnetic dynabeads^®^ M-280 streptavidin. The beads bound 133 pmol biotinylated oligonucleotides, either alone or in the presence of a 10-fold molar excess of an unlabeled normal sequence oligonucleotide (competitor). After this, SDS-PAGE-bound proteins were detected by ruthenium-staining and subsequent western blotting. The addition of a competitor displaced PARP-1 and enabled histone H1 binding. (**B**) PARP-1 or histone H1 isolated from soluble nuclear extracts of 3T3-L1 cells by immunoprecipitation bound to wild-type sequence probe (n). Binding to the fluorescent probe was inhibited by the competitor (comp). The SNV-containing fluorescent probe (1m) revealed reduced binding to PARP-1 but increased binding to histone H1 (*n* = 3, * *p* < 0.05 for comp or 1m versus n). (**C**) Western blot using immunoprecipitated histone H1 from 3T3-L1 cell soluble nuclear extract. Parylation was detected by parylation-specific antibodies. In lane 1, anti-histone H1 ab indicates the loading of an excess of the pure antibody used for immunoprecipitation. The asterisk marks the 32 kDa-band of parylated histone H1.

**Figure 4 cells-09-00427-f004:**
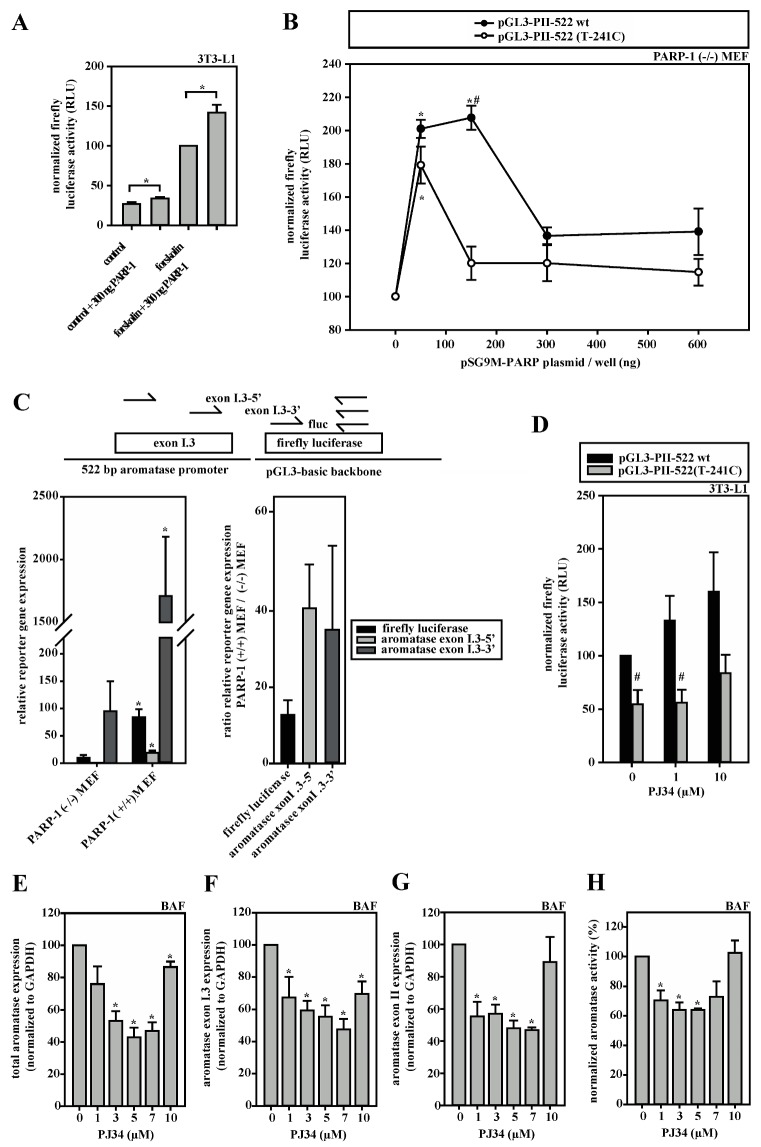
SNV-dependent aromatase expression depends on PARP-1 activity. (**A**) 3T3-L1 cells were transfected with aromatase promoter I.3/II-containing pGL3-PII-522 wt firefly luciferase reporter-plasmids and pRL-SV40. PARP-1 overexpression with 300 ng pSG9M-PARP-1 plasmid increased luciferase activity (PARP-1; *n* = 8; * *p* < 0.05). (**B**) PARP-1 (−/−) MEFs were transfected with pGL3-PII-522 wt or SNV-containing pGL3-PII-522(T-241C) plasmid. PARP-1 overexpression by co-transfection with pSG9M-PARP-1 resulted in biphasic responses of luciferase activities (*n* = 8; * *p* < 0.05; PARP-1 overexpression versus 0 ng; # *p* < 0.05 wt versus SNV-containing plasmid). (**C**) PARP-1(−/−) and PARP-1(+/+) MEFs were transfected with pGL3-PII-522 wt plasmids. Aromatase promoter usage on plasmids was measured by qRT-PCR (amplicons are indicated in the scheme; fluc, firefly luciferase), which was normalized to plasmid-DNA, measured by qPCR. The expression of luciferase including exon I.3 (5′- and 3′-regions, respectively) was similarly positively affected by PARP-1 expression (*n* = 3; * *p* < 0.05 PARP-1(+/+) versus PARP-1(−/−) MEFs). Right panel depicts expression ratios. (**D**) 3T3-L1 cells were transfected with the indicated plasmids and stimulated with forskolin. PARP-1-inhibition by PJ34 increased luciferase activity only on the wild-type-promoter I.3/II construct (*n* = 6; # *p* < 0.05 wt versus SNV-containing plasmid). (**E–H**) In forskolin-stimulated BAFs, the expression of (**E**) total aromatase, (**F**) promoter I.3- or (**G**) promoter II-containing transcripts, and (**H**) aromatase enzymatic activity were measured in the presence and absence of the PARP-1 inhibitor PJ34. Similar biphasic dose-response curves were obtained for all parameters (*n* = 3; * *p* < 0.05 versus controls without PJ34).

**Figure 5 cells-09-00427-f005:**
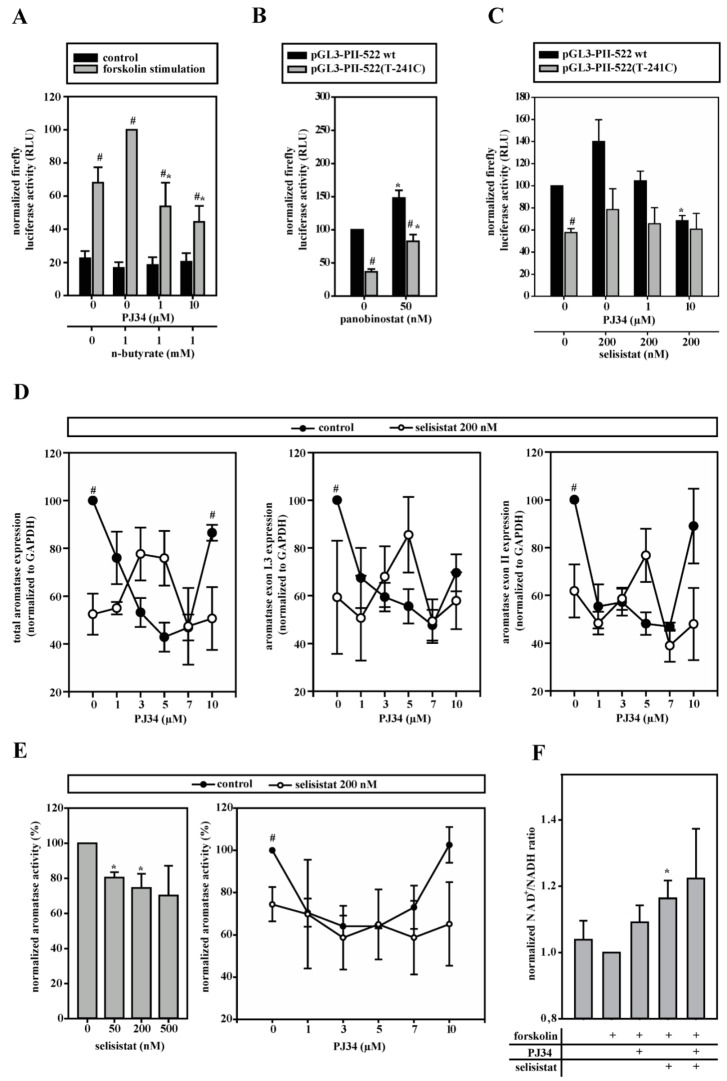
PARP-1 functionally interacts with histone deacetylases (HDACs). If not indicated otherwise, cells were stimulated with forskolin. (**A**) pGL3-PII-522 wt-transfected 3T3-L1 cells were incubated with n-butyrate, without or with PJ34 (*n* = 6; * *p* < 0.05 versus n-butyrate only; # *p* < 0.05 versus controls). (**B**) 3T3-L1 cells transfected with pGL3-PII-522 wt or pGL3-PII-522(T-241C) were incubated with panobinostat (*n* = 8; * *p* < 0.05 versus no panobinostat; # *p* < 0.05 wt versus T-241C), or (**C**) with selisistat alone or in combination with PJ34 (*n* = 8; * *p* < 0.05 versus no PJ34; # *p* < 0.05 wt versus T-241C). (**D**,**E**) Aromatase mRNA-expression or activity was measured in BAFs, which were treated with selisistat without or with PARP-1 inhibitor PJ34. For better comparison, the data from BAFs not treated with selisistat (control) were taken from Figure 4E–H (control). SIRT-1 inhibition alone reduced aromatase expression and activity. (**D**) Control *n* = 3, selisistat *n* = 6; # *p* < 0.05 control versus selisistat; (**E**) *n* = 3; * *p* < 0.05 versus no selisistat; # *p* < 0.05 control versus selisistat). (**F**) The increased NAD^+^/NADH ratios were measured in forskolin-stimulated BAFs with the inhibition of PARP-1 (5 µM PJ34) or/and SIRT-1 (200 nM selisistat) (*n* = 3; * *p* < 0.05 versus forskolin alone).

**Figure 6 cells-09-00427-f006:**
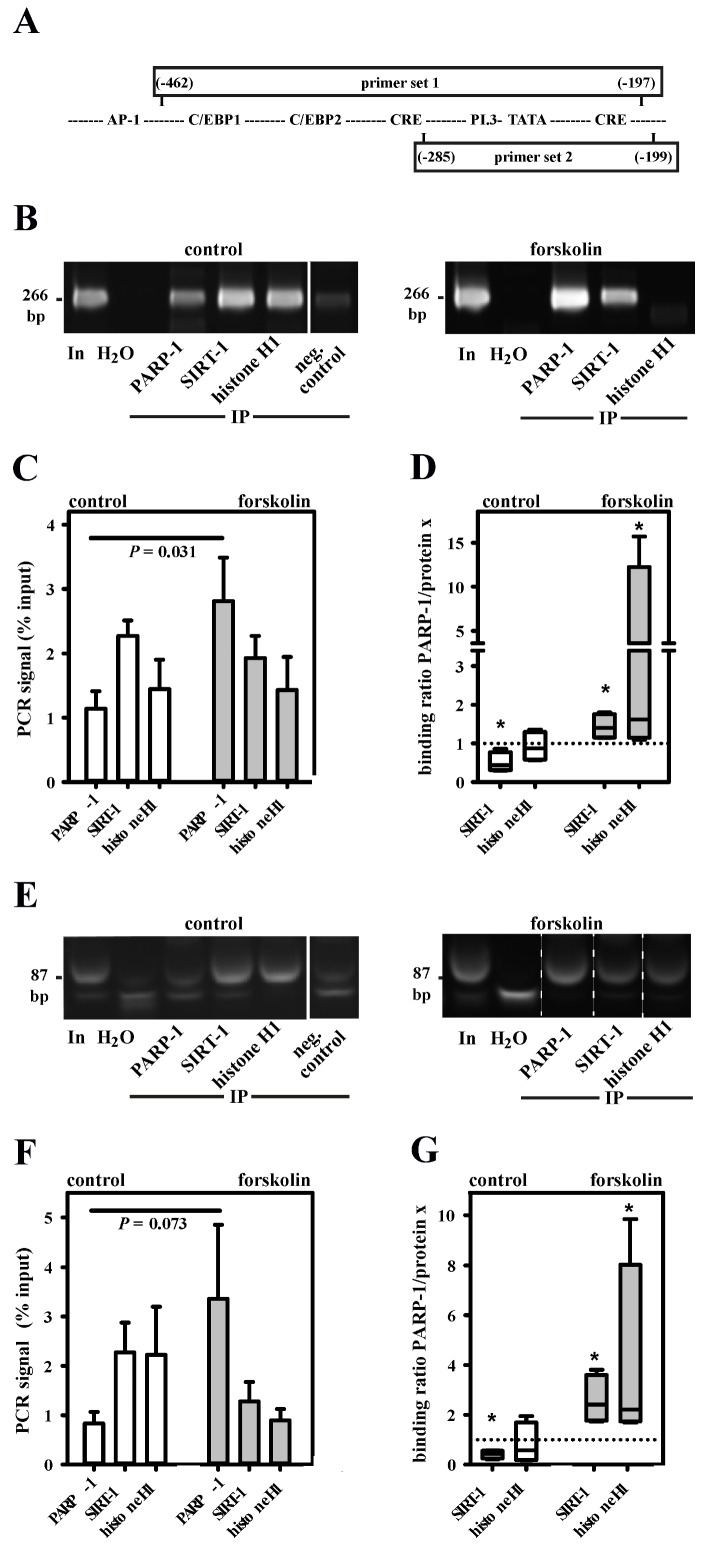
Dynamic interaction of PARP-1, SIRT-1 and histone H1 in the aromatase promoter I.3/II-region. (**A**) PARP-1, histone H1 and SIRT-1 in vivo binding to the aromatase promoter I.3/II-region in +/− forskolin-stimulated BAFs was analyzed by ChIP with two primer sets. The PCR products were separated on 12% polyacrylamide gels and stained with ethidiumbromide. In, input (1:50 diluted non-immunoprecipitated DNA); IP, immunoprecipitated samples; neg. control (protein G without antibodies). ChIP primer sets 1 ((**B**–**D**), 266 bp amplicon) and 2 ((**E**–**G**), 87 bp amplicon) revealed similar results (note that the forskolin-group image in E was cut and pasted to show the bands for PARP-1 and SIRT-1 in the general order of this figure; the original sample order is shown in Appendix A; the smaller band is unspecific). Forskolin stimulation increased PARP-1 binding and increased ratios of PARP-1/SIRT-1 and of PARP-1/histone H1 (*n* = 4; * *p* < 0.05 versus control).

**Figure 7 cells-09-00427-f007:**
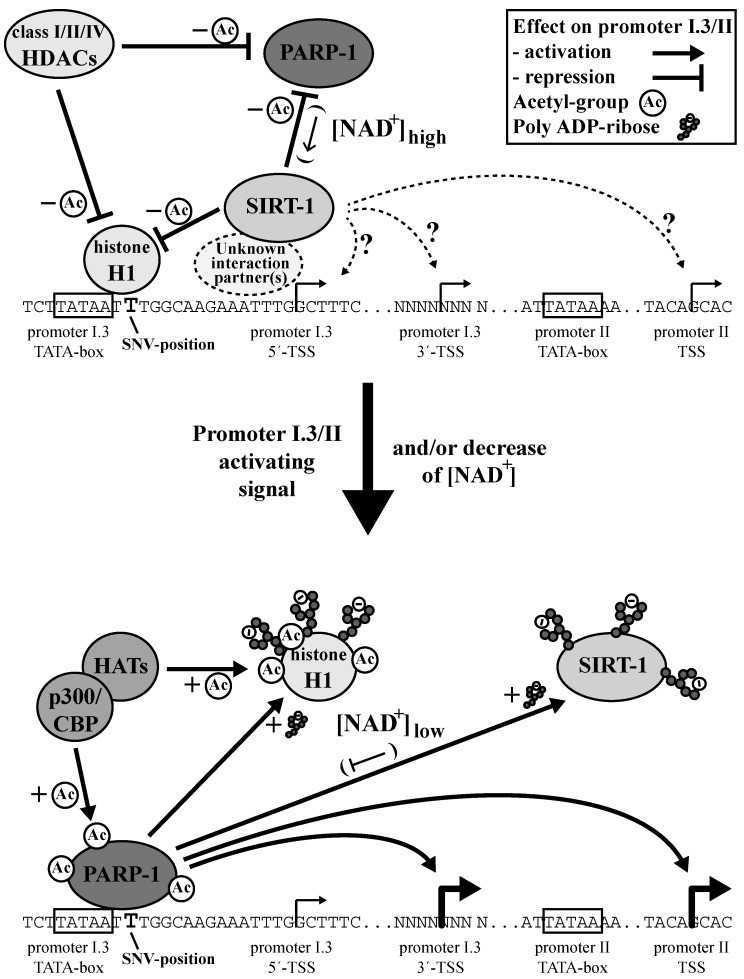
Model of PARP-1-dependent aromatase promoter I.3/II regulation. The aromatase promoter I.3/II-region is predominantly occupied by histone H1 and SIRT-1 (in concert with other HDACs) in un-stimulated BAFs. Signals activating promoter I.3/II trigger p300/CBP- or other histone acetyltransferase (HAT)-mediated acetylation of PARP-1 and H1. The resulting PARP-1 binding and activation boost displacement of H1 and SIRT-1 from the promoter by parylation. SIRT-1 activates promoter I.3/II by an unknown mechanism and, at high NAD^+^-concentrations, inhibits PARP-1 by deacetylation. Low NAD^+^-concentrations in tumor-associated BAFs favor PARP-1-activation and perpetuate H1- and SIRT-1-displacement.

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
