# Peer review of "Identification of PARP-1, Histone H1 and SIRT-1 as New Regulators of Breast Cancer-Related Aromatase Promoter I.3/II"

_cells, 2020, doi:10.3390/cells9020427_

Round 1
Reviewer 1 Report
In this manuscript, the Authors used a rare single nucleotide variant (SNV) in the cAMP-responsive aromatase promoter I.3/II-region toward the assessment of novel regulators of aromatase expression. The experimental model used allowed to identify PARP-1 and histone H1 (H1) as critical regulators of aromatase expression. PARP-1-binding to the SNV-region was crucial for aromatase promoter activation. PARP-1 parylated H1 and competed with H1 for DNA-binding, hence inhibiting its gene silencing action. The HDAC-inhibitors butyrate, panobinostat and selisistat enhanced promoter I.3/II-mediated gene expression dependent on PARP-1-activity. Forskolin-stimulation of BAFs increased promoter I.3/II-occupancy by PARP-1, whereas SIRT-1 competed with PARP-1 for DNA-binding but independently activated the promoter I.3/II. Next, the inhibition of PARP-1 and SIRT-1 increased the NAD+/NADH-ratio indicating that the NAD+/NADH-ratios may regulate the complex interactions of PARP-1, H1 and SIRT-1 and the interplay of parylation and acetylation/de-acetylation effects, therefore promoting PARP-1-activation and estrogen synthesis. On the basis of their findings, the Authors concluded that PARP-1-inhibitors may be taken into account in the treatment of estrogen-dependent breast cancers. The issue is interesting and timely, the experimental design is well performed and the manuscript is well written. However, the Authors should discuss the following references in order to strengthen their results: https://doi.org/10.3390/cells8121625 ; https://doi.org/10.3390/cells9010041
Author Response
The authors thank this reviewer for her/his critical evaluation of the manuscript and for helpful suggestions for improvements.
We revised the manuscript as detailed below.
Comment -- The Authors should discuss the following references in order to strengthen their results: https://doi.org/10.3390/cells8121625 ; https://doi.org/10.3390/cells9010041
Action – The suggested references indeed help to bring referencing of the discussion up to the current state of knowledge. We included them in the discussion.
Reviewer 2 Report
In the manuscript by Kaiser et al., the authors presented data examining the role of PARP-1 interactions with a novel single nucleotide variant on the aromatase promoter region. They further examined the competitive binding interactions of PARP-1 and Histone H1 for the SNV sequence and functional interactions of PARP-1 with SIRT-1. Based on their data, the authors conclude that PARP-1 inhibitors may be a potential inhibitor of ER+ breast cancer. Despite the value of the subject matter, the manuscript would be significantly strengthened after English language editing, as well as addressing the comments below. Based on these concerns, I recommend that the manuscript require revisions before acceptance to this journal.
Comments:
Some language proofing/editing is needed to improve the overall clarity and accuracy of the manuscript. An initial figure/schematic diagram giving an overview of the potential regulatory mechanisms and interactions to be examined in the results section may be useful. Figure 1B: The inhibition of Complex 1 formation by the competitor oligonucleotide appears to be attenuated by forskolin. Please discuss this observation. Figure 2C: The formation of Complex 1 is very difficult to observe in the image of the gel provided. Figure 2D: If possible, please provide a loading control for this gel. Figure 6: Discuss the results/significance from each panel individually as you did for previous figures. Also, what is the reasoning for the “reverse order of PARP-1 and SIRT-1 on the forskolin gel in E”. The order should be consistent to avoid misinterpretation of the data. Preliminary data testing the potential inhibitory effects of PARP-1 inhibitors on ER+ breast cancer cell lines would significantly strengthen the authors’ conclusions.Author Response
The authors thank this reviewer for her/his critical evaluation of the manuscript and for helpful suggestions for improvements.
We revised the manuscript as detailed below.
Comment -- Some language proofing/editing is needed to improve the overall clarity and accuracy of the manuscript.
Action -- The manuscript was language edited (the language editor is acknowledged in the manuscript).
Comment -- An initial figure/schematic diagram giving an overview of the potential regulatory mechanisms and interactions to be examined in the results section may be useful.
Action: -- A flow chart highlighting the main questions addressed in the study was added as Figure 1C (the location was choosen mainly as a result of layout considerations). We refrained from adding a more detailed figure at this point, because the manuscript already contains Figure 7. We added a reference to Figure 7 to the legend to Figure 1 (if the editor(s) permits that).
Comment -- Figure 1B: The inhibition of Complex 1 formation by the competitor oligonucleotide appears to be attenuated by forskolin. Please discuss this observation.
Action: -- We added the quantitative evaluation of 6 experiments (“n=4” in the manuscript was a copy-paste error). It shows that in summary this observation is not significantly different from the complex 1 signals obtained with the mutated probes. Therefore, we refrained from discussing this explicitly because we think that the quantitation data put that band in the right context. We show this respective gel, because it contains all conditions in the logical order appearing in the text and contains no superfluous conditions.
Comment -- Figure 2C: The formation of Complex 1 is very difficult to observe in the image of the gel provided.
Action: -- We agree at least partially with the reviewer concerning Figure 2C. However, for the same reasons outlined above we would keep this gel. To get a better visibility of the complex 1-band we will provide the original illustrator-file for the editor(s) to include the highest resolution image in the finalized manuscript.
Comment -- Figure 2D: If possible, please provide a loading control for this gel.
Action: -- Concerning Figure 2D we did not do a loading control and therefore cannot provide it. We thought this would be dispensable at this point, because the Figure 2D is intended only to reassure the reader that PARP-1 protein is absent in the (-/-) MEFs, which we obtained as noted in the Methods and Acknowledgement sections, respectively, directly from Dr. Wang, which generated them (Wang ZQ et al.. 1995. Mice lacking ADPRT and poly(ADP-ribosyl)ation develop normally but are susceptible to skin disease. Genes & Development, 9 (5):509-520.).
Comment -- Figure 6: Discuss the results/significance from each panel individually as you did for previous figures. Also, what is the reasoning for the “reverse order of PARP-1 and SIRT-1 on the forskolin gel in E”. The order should be consistent to avoid misinterpretation of the data.
Action: -- As suggested by the reviewer we now discuss the results from each panel individually. However, we combined panels B+C and E+F, respectively, as in each pair of panels one panel shows original gels and the other presents the summarized data for all independent experiments.
The reasoning for the “reverse order” originated from an error in the gel loading plan for the experiments with primer set 2 (forskolin samples only); for consistency in the lab notes, we kept that loading scheme for all replicate experiments. To provide a consistent sample order for the readers, we now use a corrected (“cut-paste”) image for Figure 6E and include the original image in the Supplementary Materials.
Comment -- Preliminary data testing the potential inhibitory effects of PARP-1 inhibitors on ER+ breast cancer cell lines would significantly strengthen the authors’ conclusions.
Action: -- In principle we agree with the reviewer. However, we hope that the reviewer accepts that this would be a major additional effort, worth to be done in future work.
As almost all ER+ breast cancer cells do not express enough aromatase to support their own growth, they are dependent on systemic supply or BAFs, which express high levels of aromatase. This lack of (sufficient) own aromatase expression makes most ER+ breast cancer cell lines dependent on estrogen added to the medium. Concerning the proposed experiment(s), this would require co-cultures of breast cancer cell lines with BAFs or working with BAF-conditioned media on the cancer cell lines. Both systems would require a considerable amount of setup-time to be conducted properly.
Reviewer 3 Report
In this manuscript, A. Kaiser and coworkers identified a rare single variant in aromatase promoter which reduces its activity. Furthermore, they proposed the mechanism involved in this reduction. Thus PARP-1 seems to be a key regulator in this promoter activity.
The authors first discussed the mechanism of action and proposed the use of PARP-1 inhibitors as a new treatment aproach to reduce aromatase expression in peritumoral tissue, not only in ER-negative breast cancers but also in ER-positive.
The overall study is well organized and thus I support acceptance of this work.
Author Response
The authors thank this reviewer for her/his critical evaluation of the manuscript.
Round 2
Reviewer 1 Report
The Authors addressed the Reviewer's suggestion.